# Fluidity onset in graphene

Denis A. Bandurin[1], Andrey V. Shytov[2], Leonid S. Levitov[3], Roshan Krishna Kumar[1,4], Alexey I. Berdyugin[1], Moshe Ben Shalom[1,4], Irina V. Grigorieva [1], Andre K. Geim[1,4] & Gregory Falkovich[5,6]

Viscous electron fluids have emerged recently as a new paradigm of strongly-correlated electron transport in solids. Here we report on a direct observation of the transition to this long-sought-for state of matter in a high-mobility electron system in graphene. Unexpectedly, the electron flow is found to be interaction-dominated but non-hydrodynamic (quasiballistic) in a wide temperature range, showing signatures of viscous flows only at relatively high temperatures. The transition between the two regimes is characterized by a sharp maximum of negative resistance, probed in proximity to the current injector. The resistance decreases as the system goes deeper into the hydrodynamic regime. In a perfect darkness-before-daybreak manner, the interaction-dominated negative response is strongest at the transition to the quasiballistic regime. Our work provides the first demonstration of how the viscous fluid behavior emerges in an interacting electron system.

[1] School of Physics, University of Manchester, Oxford Road, Manchester M13 9PL, UK. [2] School of Physics, University of Exeter, Stocker Road, Exeter EX4 4QL, UK. [3] Department of Physics, Massachusetts Institute of Technology, 77 Massachusetts Ave, Cambridge MA02139, USA. [4] National Graphene Institute, University of Manchester, Manchester M13 9PL, UK. [5] Weizmann Institute of Science, Rehovot, Israel. [6] Novosibirsk State University, Novosibirsk, Russia 630090. These authors contributed equally: Denis A. Bandurin, Andrey V. Shytov. Correspondence and requests for materials should be addressed to G.F. (email: gregory.falkovich@weizmann.ac.il)

Electron fluids, an exotic state of matter in which electron–electron (ee) interactions dominate transport, have been long anticipated theoretically[1–15] but until recently they were far from experimental reality. This situation is currently changing owing to the discovery of new materials in which ee interactions are particularly strong or momentum relaxation due to disorder and phonons is weak. The inventory of experimental systems that can host viscous e-fluids, as we will call them for brevity, has been steadily growing in the last few years[16–19], stimulating wide interest in their properties. E-fluids may exhibit new behaviors such as vortices[20,21], whirlpools[16], superballistic transport[22,23], Poiseuille flow[10,11,13,14,18], anomalous heat conduction[17], and viscous magnetotransport[24,25]. The questions about the genesis of e-fluids, on the other hand, received relatively little attention. How does an electron system enter the fluid state? What happens when $l_{ee}$ becomes comparable or larger than the system dimensions? What is the relation between electric current and potential at the transition? All these questions are at present poorly understood: neither there exists a detailed theory treating both ballistic and viscous electron regimes on equal footing, nor any systematic experimental study of the transition has been performed. Searching for the fluidity onset is the subject of this work.

So far, the behavior of e-fluids was mostly discussed deep in the hydrodynamic regime, where the mean free path $l_{ee}$ was the shortest lengthscale of the system. However, the experimental conditions are usually such that $l_{ee}$, tunable by varying temperature $T$, is either comparable or at most a few times smaller than the system dimensions, putting the experimentally investigated e-fluids close to the onset of fluidity. As we will show below, this regime hosts an interaction-dominated quasiballistic state, which exhibits a negative voltage response similar to that observed at not-too-high $T$ in the ref.[16]. The negative response arises because ambient carriers, as a result of momentum-conserving collisions with injected carriers, are blocked from reaching voltage probes. Furthermore, the negative response is enhanced by "memory effects", so that it may exceed the negative response in the viscous state[26]. Thus, the interaction-dominated quasiballistic state, while quite distinct from the viscous fluid state, can in some cases serve as a proxy for the latter.

Graphene offers a convenient venue for this study. First, due to their exceptional cleanness and weak electron–phonon (el–ph) coupling, state-of-the-art graphene devices support micrometer-scale ballistic transport with respect to momentum-non-conserving collisions over a wide range of temperatures[27], from liquid-helium to room $T$. Second, above the temperatures of liquid nitrogen, ee collisions become the dominant scattering mechanism, so that the behavior of the electron system resembles that of viscous fluids[16,23]. Third, $l_{ee}$ in graphene can be varied over a wide range[23] by changing the carrier density $n$ and $T$. This enables a smooth transition (or, more precisely, a crossover) between single-particle ballistic and viscous transport regimes, allowing us to track how the electron system enters the collective fluid state.

## Results

**Experimental data.** We explore the onset of the hydrodynamic state by studying graphene devices in the so-called vicinity geometry[16], illustrated in Fig. 1a: The current $I$ is injected through a narrow contact into a wide graphene channel, and a local potential is probed at a small distance $x$ from the injector. The main result of our study is that the vicinity resistance $R_v = V/I$ reaches an extreme negative value at the onset of fluidity. In particular, this behavior manifests itself most clearly through the temperature dependence of $R_v$ (Fig. 1b, c), with the quasiballistic

and hydrodynamic regimes occurring at low and high $T$, respectively. We will show that the deep minimum at intermediate temperatures in the $R_v(T)$ dependences is the hallmark of the transition. Furthermore, we will demonstrate that this transition can be conveniently quantified by the electron Knudsen number

$$\mathrm{Kn} = l_{ee}/x, \tag{1}$$

taking values $\mathrm{Kn} \ll 1$ and $\mathrm{Kn} > 1$ in the hydrodynamic and quasiballistic transport regimes, respectively, and approaching unity at the fluidity onset.

Importantly, the negative sign of $R_v$, observed across the entire transition, signals that ee interactions dominate transport in both the quasiballistic and hydrodynamic regimes. The hydrodynamic regime, where theory predicts $dR_v/dT > 0$[20,28], occurs only at high enough temperatures and low enough carrier densities. This regime is preceded by an extended quasiballistic regime with $dR_v/dT < 0$, discussed in detail below. The occurrence of two distinct interaction-dominated regimes in a 2D electron system is a surprising finding, which is of interest from a fundamental perspective and important for possible applications.

To explore the onset of the fluid state experimentally, we fabricated high-quality devices based on bilayer graphene (BLG) encapsulated between hexagonal boron nitride (for details, see Methods). The latter provides a clean environment for graphene's electron system ensuring micrometer-scale ballistic transport with respect to extrinsic momentum-non-conserving scattering. The

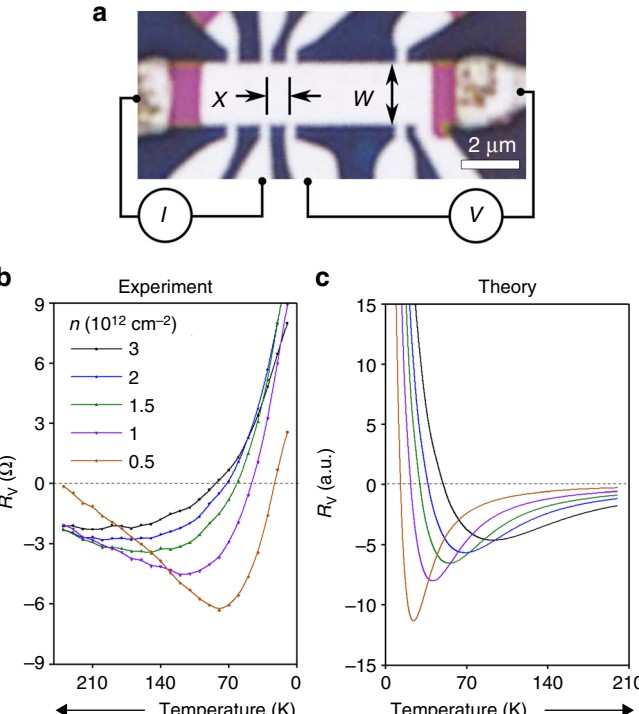

**Fig. 1** Vicinity resistance $R_v$. **a** Optical photograph of one of our devices on which the measurement geometry is indicated: current $I$ is injected into the graphene channel through a 300 nm contact and the voltage drop is measured at a distance $x$ from the injection point. Device width $W$ is 2.3 μm. **b, c** Temperature dependence of the vicinity resistance measured experimentally and computed theoretically for different carrier densities $n$ in bilayer graphene. The most negative value occurs at the fluidity onset, $\mathrm{Kn} \sim 1$, where $\mathrm{Kn}$ is the Knudsen number, (1)

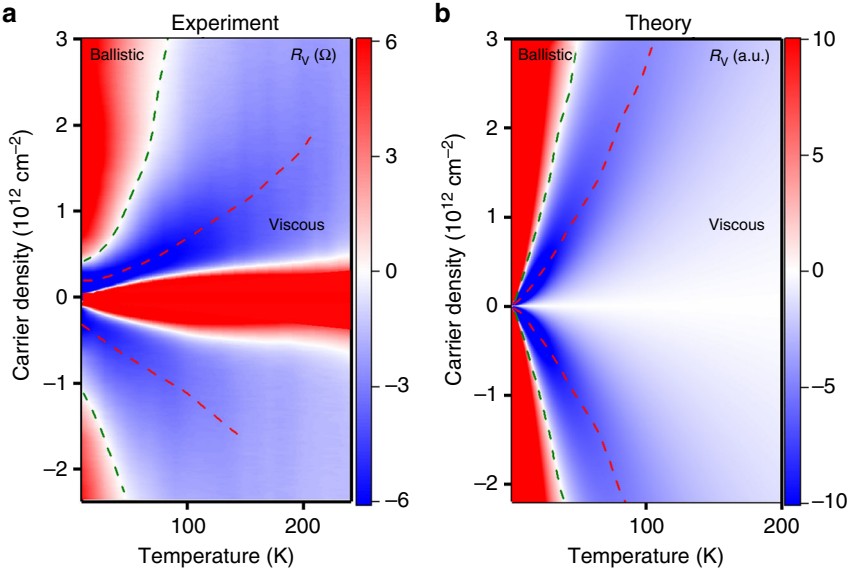

**Fig. 2** Vicinity resistance $R_v$ as a function of carrier density and temperature. The dashed green line indicates zero resistance. Dashed red lines: minima in the resistance. **a** Experiment: $R_v(n,T)$ for bilayer graphene. The central red region indicates the density range around the CNP where our hydrodynamic analysis is inapplicable. **b** Theory: resistance obtained by solving the kinetic equation. The key features in both panels: sign reversal at quasiballistic-to-hydrodynamic transition, the maximal negative signal at the onset of the viscous regime, and a slow decay of the signal at higher temperatures

devices were shaped in a form of dual-gated multiterminal Hall bars (Fig. 1a), allowing us to study the distance-dependent potential anticipated at the transition upon varying the carrier densities $n$. The dual-gated design allowed us to maintain zero displacement between the graphene layers, so that one could tune the Fermi energy $\varepsilon_F$ in BLG without altering its band structure (opening the band gap). We have strategically chosen the BLG system because it $\varepsilon_F$ varies with $n$ stronger than in monolayer graphene (MLG) ($n$ vs. $n^{1/2}$). The standard dependence $l_{ee} = \hbar v_F \varepsilon_F/(k_B T)^2$ translates into the scaling $l_{ee} \sim n^{3/2}$, which is much faster than the $n^{1/2}$ dependence in MLG. This allowed us to explore a wider range of $l_{ee}$ than in MLG by varying the carrier density for a given $T$ (see below), providing a convenient knob to tune the Kn value and probe the quasiballistic-to-hydrodynamic transition[29].

Notably, the signal measured in the vicinity configuration contains a non-negligible offset due to momentum-non-conserving scattering (by phonons and/or disorder) which we further refer to as an Ohmic contribution. To distill the viscous contribution, we employed the approach introduced in the ref. [16] in which the Ohmic term, expressed as $b\rho$, was subtracted from the measured vicinity signal, assuming the additive behavior of these contributions[28]. Here $\rho = \rho(n, T)$ is the BLG sheet resistance measured in the conventional four-terminal geometry and $b$ is the geometric factor that depends on sample dimensions and the distance between the injection point and the voltage probe[16,28] (for example, $b = 0.1$ for the measurement configuration shown in Fig. 1a). As discussed below, the procedure of subtracting the Ohmic contribution, while somewhat ad hoc, can be justified for the geometry of our experiment. Below we refer to this adjusted vicinity resistance using the same notation $R_v$ unless stated otherwise.

Figure 1b shows $R_v$ as a function of $T$ measured in one of our BLG devices. Far away from the charge neutrality point (CNP) and at liquid helium $T$, $R_v$ is positive for all experimentally accessible $n$. When the temperature is increased, $R_v$ rapidly drops, reverses its sign, reaches a minimum and then starts to grow. Figure 2a details this observation by mapping $R_v$ on the $(n, T)$-plane. The non-monotonic dependence $R_v$ vs. $T$ is observed for all

$n$, whereby the temperature at which $R_v$ dips, grows with increasing $n$ (red dashed line).

To understand this nonmonotonic behavior, we first consider the limiting cases: the hydrodynamic regime $l_{ee} \ll x$, realized at large $T$, and the free-particle regime $l_{ee} \gg W$, realized at the lowest $T$ (here $W$ is the device width). In the hydrodynamic regime, negative $R_v$ arises as a result of viscous entrainment by the injected current of the fluid in adjacent regions[16,20,28]. In the free-particle regime, positive $R_v$ is expected from single-particle ballistic transport due to reflection of injected carriers from the opposite boundaries[30,31]. Therefore, the sign of $R_v$ must change from negative to positive upon lowering $T$, as indeed seen in the data shown in Fig. 1a. Furthermore, the hydrodynamic $R_v$ is proportional to viscosity[20,28], giving the dependence $R_v \sim l_{ee}(T)$. The quantity $l_{ee}(T)$ increases as $T$ decreases, leading to increasingly more negative $R_v$. The non-monotonic temperature dependence $R_v(T)$, implied by these observations, is indeed seen in our measurements (Figs. 1b, 2a).

Importantly, in between the free-particle regime $l_{ee} \gg W$ and the hydrodynamic regime $l_{ee} \gg x$ lies an interesting regime $x < l_{ee} < W$ that has hitherto been ignored in the literature. This intermediate regime, which for the lack of a better name will be called "quasiballistic", features an interaction-dominated response of a non-hydrodynamic nature, since the mean free path $l_{ee}$ is greater than the distance from the injector to the probe. Conspicuously, $R_v$ remains negative in this regime. However, since now $R_v \sim 1/l_{ee}(T)$, the sign of $dR_v/dT$ is reversed compared to the hydrodynamic regime. The negative sign of $R_v$ can be understood by considering injected carriers that travel over a large distance of the order of $l_{ee} > x$ and then scatter off ambient thermal carriers. After scattering, some of the injected carriers make it back into the probe, creating a positive contribution to $R_v$. Simultaneously, some of the ambient carriers, through scattering off the injected carriers, are blocked from reaching the probe. This process creates a negative contribution to $R_v$. Detailed analysis shows that the latter contribution dominates[26], giving rise to negative $R_v$. As $T$ increases, $R_v$ grows progressively more negative until the point $l_{ee} = x$, where the hydrodynamic behavior sets in and the sign of the $T$ dependence is reversed. Interestingly, in the quasiballistic regime,

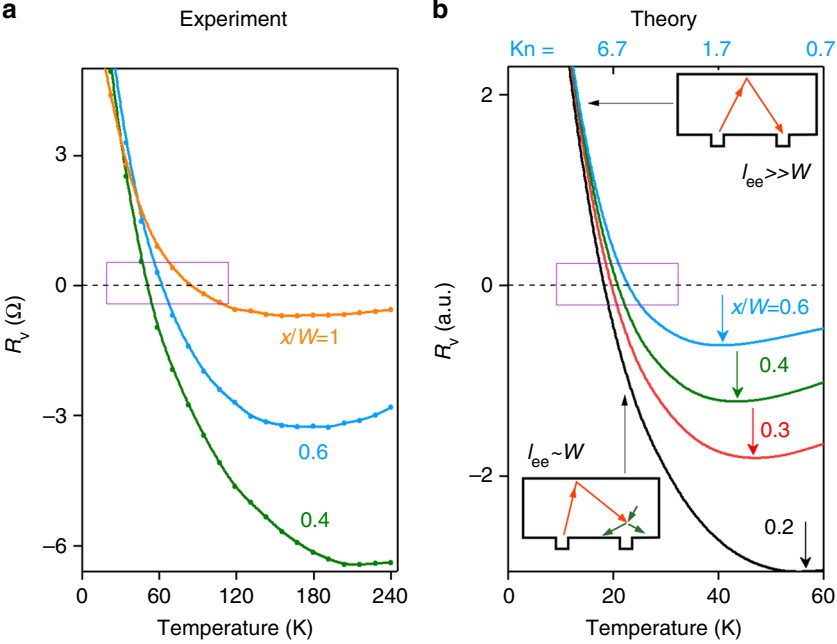

**Fig. 3** Vicinity resistance versus $T$ for several positions $x$ of the voltage probe. **a** Experimental $R_v(T)$ for $n = 1.5 \times 10^{12}$ cm$^{-2}$ and $W = 2.3$ μm. **b** Theoretical dependence $R_v(T)$ predicted from the kinetic equation for the device in Fig. 1a. Temperature enters through the ee scattering rate $\hbar\gamma_{ee} \simeq T_e^2/\varepsilon_F$[17]. The top axis corresponds to the electron Knudsen number Kn $= l_{ee}/x$ calculated for the blue curve. The purple rectangles and arrows mark, respectively, the sign change and minima in the $R_v(T)$ dependences. Upper inset: Schematics of electron transport in the ballistic regime. The potential at the source is transported by carriers throughout the sample and into the probe. Lower inset: Schematics of the voltage sign change in the quasiballistic regime. Collisions between injected carriers (red arrows) and ambient thermal carriers (green arrows) diminish the number of thermal carriers reaching the probe. This reduces the potential, which reverses its sign and becomes negative. Arrows link the insets with the corresponding portions of the $R_v(T)$ dependence

the value $|R_v|$ decreases with $n$ and grows with $T$, in qualitative agreement with the behavior of a MLG $R_v$ at not-too-high $T$ found in the ref. [16]. This suggests a possible resolution of the conundrum posed by the findings of ref. [16], in which a hydrodynamic-like negative $R_v$ was found to depend on $n$ and $T$ differently from what is expected in the hydrodynamic regime.

**Theory and comparison with experiment.** To capture all these different regimes in a single model, we employ the kinetic equation for quasiparticles in the graphene Fermi liquid. Transport in the geometry of Fig. 1 is described by solving the kinetic equation in an infinite strip of width $W$: $-\infty < x < \infty$, $0 < y < W$, with diffuse boundary conditions at the strip edges $y = 0, W$. Current $I$ is injected through a point-like source at $x = y = 0$ and is drained on the far left, $x = -\infty$. We find the potential at $(x, 0)$ by evaluating the particle flux entering the probe (for details, see Methods). At low temperatures the ee rate $\gamma_{ee}$ is small, and the ee collision term can be ignored[12]. The model then describes ballistic particles bouncing between the strip edges, as illustrated in the upper inset of Fig. 3b. The net flux of particles into the probe then gives a positive value $R_v = V_p(x)/I$. At high $T$, on the other hand, the ee collision term dominates, and the distribution function approaches the local equilibrium. The resulting hydrodynamic behavior is then described by the Stokes equation that states the balance between the viscous friction and electric forces: $e\mathbf{E}/m = -\nu\nabla^2\mathbf{v}$. (The latter follows directly from the Eq. (12) of Methods, multiplied by $\mathbf{p}$, integrated over momenta and combined with an expression for the stress tensor obtained from $1/\gamma_{ee}$ expansion.) In this case, we obtain $R_v \sim \eta/(nex)^2$, where $\eta$ is the dynamic viscosity given by $\eta = \frac{1}{4}m^*n v_F l_{ee}$ and $m^*$ is the carrier effective mass[20,22]. The single parameter $\gamma_{ee}$

allows us to explore both the ballistic and viscous regime through the dependence of $R_v$ on $T$ and $n$. Carrier dynamics in the quasiballistic regime is shown schematically in the lower inset of Fig. 3b.

In Fig. 1b, c we compare the experimental data for $R_v$ vs. $T$ with the results of our modeling, assuming the ee collision rate that depends on $T$ and $n$ as $\hbar\gamma_{ee} \sim T_e^2/\varepsilon_F$ [12]. For bilayer graphene, the Fermi energy $\varepsilon_F$ is related to the carrier density as $n = m^*\varepsilon_F/(\pi\hbar^2)$, where $m^* = 0.033\, m_e$. The two panels flaunt good qualitative agreement; namely, our theory captures the main experimental features: positive $R_v$ at small $T$ that rapidly drops with increasing $T$ and monotonically grows with $n$, so that the minima and sign changes in $R_v$ occur at higher $T$ for larger $n$.

Furthermore, our model reproduces some of the more subtle features of the data. For example, the nodes in $R_v$ vs. $T$ shift to higher $T$ and the minima to lower $T$, as the distance to the probe $x$ increases, see Fig. 3. An overall agreement is also found for the full $R_v(n, T)$ maps shown in Fig. 2a, b that become near-identical after rescaling the $T$ axis.

## Discussion

In our analysis, for simplicity, we disregarded the Ohmic effects due to the el–ph scattering. This is a reasonable starting point since the el–ph scattering mean free path $l_{el-ph}$ is considerably larger than $l_{ee}$ at the temperatures of interest (for details, see Methods). However, the flow can be distorted by the Ohmic effects at the lengthscales set by $\xi = \sqrt{\eta/n^2e^2\rho} = \frac{1}{2}\sqrt{l_{el-ph}l_{ee}}$, which lies between $l_{ee}$ and $l_{el-ph}$[20,21]. Thus caution must be exercised even when the el–ph scattering is weak. The procedure of extracting the viscous contribution by subtracting the Ohmic contribution is expected to work well so long as the Ohmic effects

do not distort the current flow at the lengthscales which are being probed, i.e. when $\xi$ exceeds the distance to the probe $x \approx 1\,\mu m$. Estimates show that the inequality $\xi \gg x$ holds at not-too-high temperatures, i.e. in the quasiballistic regime. At the fluidity onset, identified above as the turning point in the $R_v(T)$ dependence, for the estimated typical values $l_{ee} \lesssim 0.2\,\mu m$ and $l_{el-ph} \sim 3\,\mu m$, the lengthscale $\xi$ can become comparable to $x$. However, an analysis based on the Stokes equation indicates that, for the geometry of our experiment, the Ohmic and viscous contributions remain approximately additive even for $\xi < x$ (for details, see Methods). We therefore believe that the subtraction procedure provides a reasonable approximation in the entire range of temperatures and dopings.

We also note that Figs. 2, 3 exhibit some discrepancy between the values of $T$ at which theoretical and experimental $R_v$ reach the minimum. This is not particularly surprising given the simplistic expression of $\gamma_{ee} \sim T^2$ used in the model. Since $\gamma_{ee}$ is the only relevant temperature-dependent parameter in the model, the quantitative agreement can be improved through revising the dependence $\gamma_{ee}$ vs. $T$. Indeed, there are various effects that can give rise to deviation from the standard Fermi-liquid $T^2$ dependence. One is the logarithmic enhancement of the quasiparticle decay rate due to collinear ee collisions[32–34]. However, it is probably an unlikely culprit, since collinear collisions do not lead to angular relaxation. At the same time, recent analysis[35] indicates that the effective $\gamma_{ee}$ that determines electron viscosity depends on the lifetimes of the odd-$m$ angular harmonics, $m = \pm 3, \pm 5,...$, which relax considerably slower than the Fermi-liquid $T^2$ estimate would suggest. Accounting for this effect could, effectively, extend the quasiballistic behavior to higher temperatures, which would improve the agreement with the observed dependence $R_v(T)$. Detailed analysis of these rates and of their impact on $R_v$ is beyond the scope of this work.

The experimental and theoretical $R_v(T)$ exhibit two prominent features: $R_v$ first changes sign from positive to negative and then passes through a deep minimum. Should the sign change or the minimum be taken as the signature of the onset of fluidity? That question can be answered with the help of the data presented in Fig. 3, demonstrating that $R_v$ is a non-trivial function of both $l_{ee}/W$ and $l_{ee}/x$. We note in that regard that the sign reversal of theoretically computed $R_v$ occurs at Kn $\gg 1$, that is inside the quasiballistic regime, for all values of $x$ (Fig. 3b). Indeed, $R_v$ in Fig. 3b changes sign at $T = 20\,K$ which for a given $n$ translates into $l_{ee} \approx 10\,\mu m$, a length scale significantly greater than the values $x \sim 1 - 2\,\mu m$ for this device. On the other hand, the most negative $R_v$ in Fig. 3b is found at Kn $= 1-3$, which corresponds to $x \sim l_{ee} < W$. Since in the hydrodynamic regime $R_v$ is proportional to $\eta$ and thus should drop with increasing $T$, we infer that it is the condition Kn $\sim 1$ (where $R_v$ is most negative) that describes the fluidity onset. Furthermore, $R_v$ is expected to be negative in the quasiballistic regime[26] when Kn $> 1$, so it is indeed the drop of $|R_v|$ with temperature, rather than the sign reversal, that marks the onset of the viscous flow.

Experimental observation of this anomalous behavior at the onset of the fluid state enables a direct electrical measurement of the mean free path $l_{ee}$ and electron viscosity. Good qualitative agreement of the experimental data and our theoretical model suggests further opportunities to study the physics of e-fluids, in particular the electron transport in the presence of magnetic field and/or confining potential, obstacles, funnels and electron pumps. Our work clearly shows that the initial deviation from the ballistic behavior observed experimentally in different systems[13,14,16,18,23] may be due to an entry into the interaction-dominated "quasiballistic" regime rather than the true onset of electron fluidity. It requires higher temperatures and the

observation of the behavior consistent with viscosity gradually decreasing with increasing $T$ to ascertain that the Navier-Stokes description can be applied.

## Methods

**Device fabrication**. Our devices were made of bilayer graphene encapsulated between $\approx 50\,nm$-thick crystals of hexagonal boron nitride (hBN). The hBN-graphene-hBN heterostructures were assembled using the dry-peel technique described elsewhere[27,36] and deposited on top of an oxidized Si wafer (290 nm of SiO$_2$) which served as a back gate. After this, a PMMA mask was fabricated on top of the hBN-graphene-hBN stack by electron-beam lithography. This mask was used to define contact areas to graphene, which was done by dry etching with fast selective removal of hBN[37]. Metallic contacts (usually, 5 nm of chromium followed by 50 nm gold) were then deposited onto exposed graphene edges that were a few nm wide. As the next step, another round of electron-beam lithography was used to prepare a thin metallic mask (40 nm Al) which defined a multiterminal Hall bar. After this, reactive ion plasma etching translated the shape of the metallic mask into encapsulated graphene. The Al mask also served as a top gate, in which case Al was wet-etched near the contact leads to remove the electrical contact to graphene.

**Distilling the hydrodynamic contribution in the presence of Ohmic effects**. Here we assess the accuracy of the approach used in the main text to separate the viscous and Ohmic contributions to the $R_v$ signal. In this approach, it was assumed that the contributions are approximately additive, and thus the viscous contribution can be distilled by subtracting the (suitably scaled) Ohmic resistivity measured in a four-probe setup.

The validity of the additivity assumption can be verified using an exact solution of the hydrodynamic equations for current injected in a halfplane. The hydrodynamic approach applies when the ee mean free path is smaller than the el-ph scattering mean free path, $l_{ee} \ll l_{el-ph}$. At the scales larger than $l_{ee}$ the electron flow satisfies the Stokes equation with an Ohmic term added to describe momentum relaxation:

$$\left(\eta \nabla^2 - n^2 e^2 \rho\right)\mathbf{v} = ne\nabla\phi \tag{2}$$

Taking a curl and defining $\kappa^2 = \rho(en)^2/\eta = 1/\xi^2$, we obtain the equation on the stream function:

$$\left(\nabla^4 - \kappa^2\nabla^2\right)\psi = 0, \tag{3}$$

where $v = \nabla \times (\psi\mathbf{z})$. Following[21], we consider the flow in a half-plane $y > 0$ generated by the a point source on the boundary at $x = 0$: $\psi_x(x, 0) = \delta(x)I/ne$. The stream function in this case has the form

$$\psi(x,y) = \frac{I}{ne}\int\frac{e^{ikx}dk}{2\pi ik}\left[Ae^{-|k|y} + (1-A)e^{-qy}\right], \tag{4}$$

where we defined $q = \sqrt{k^2 + \kappa^2} > 0$. The stream function can be used to evaluate the potential. Plugging Eq. (4) in Eq. (2), we see that only the first (harmonic) term in the stream function contributes the potential:

$$\nabla\phi = \frac{\eta}{en}(\nabla^2 - \kappa^2)\mathbf{v} = \frac{\eta}{en}(\nabla^2 - \kappa^2)\mathbf{z}\times\nabla\psi = -\frac{\eta I\kappa^2}{2\pi(en)^2}\mathbf{z}\times\nabla\int\frac{Ae^{ikx-|k|y}dk}{ik}. \tag{5}$$

The yet-undetermined quantity $A(k)$ depends on the type of boundary condition. The no-stress boundary condition at $y = 0$, which reads $\psi_{yy}(x, 0) = 0$, yields $A(k) = q^2/\kappa^2 = 1 + k^2/\kappa^2$. Remarkably, the exact potential is a sum of the viscous and Ohmic contributions, with each contribution unaffected by the presence of the other contribution in this case:

$$\phi(x,y) = \frac{I}{2\pi}\left[\frac{2\eta}{(en)^2}y^2 - x^2(x^2 + y^2)^2 + \rho\log\left(\frac{L^2}{x^2+y^2}\right)\right], \tag{6}$$

where $L$ is the system size. The subtraction procedure employed in analyzing the measurements is exact at all distances for the no-stress boundary condition.

For the no-slip boundary condition, on the other hand, the additivity is only an approximate property. In this case, $\psi_y(x, 0) = 0$ gives

$$A(k) = 1 + k^2/\kappa^2 + q|k|/\kappa^2. \tag{7}$$

The last term in this expression gives a contribution which depends both on viscosity and resistivity. As illustrated in Fig. 4, this contribution is non-negligible at distances $x \simeq \xi$, where $R_v$ changes sign. Its magnitude, however, is small (under 10–15% of the total potential). Therefore, disregarding this contribution should

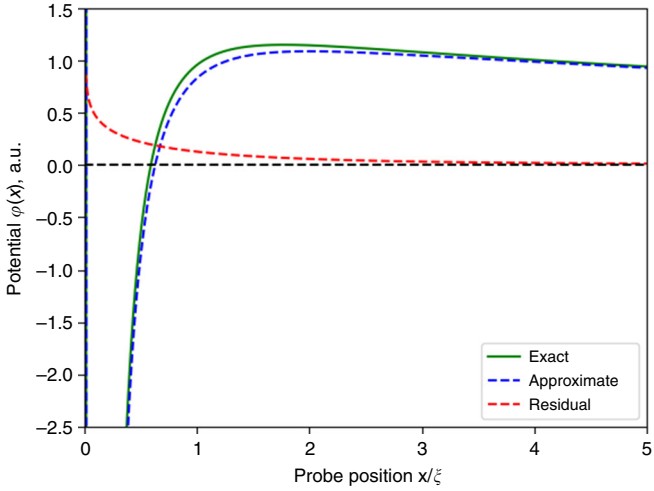

**Fig. 4** Potential at the edge of a halfplane as a function of the distance to the current injector, calculated for the no-slip boundary conditions. The exact result (green curve) can be approximated by a direct sum of the Ohmic and viscous contributions (dashed blue curve). The residual (dashed red curve) is the non-additive part, defined as the difference of the exact and approximate potentials. At $x \sim \xi$ the residual constitutes no more than 10–15% of the net potential value, becoming much smaller at $x \gg \xi$ and $x \ll \xi$

provide a reasonably good approximation. Yet, this conclusion is almost certainly geometry-sensitive, being valid for the point source at a halfplane edge but not necessarily for other geometries.

**Estimates of the electron-phonon scattering mean free path.** Electron-phonon scattering rate in graphene was discussed mostly for the single-layer case[38–40]. Here we modify this analysis for the bilayer case. The value of the mean free path $l_{el-ph}$ is used in the main text to determine the lengthscales at which the el-ph scattering does not distort the carrier flow.

We use the standard deformation potential Hamiltonian

$$H_{el-ph} = \int d^2 r \psi^\dagger(\mathbf{r}, t) D \nabla \mathbf{u}(\mathbf{r}, t) \psi(\mathbf{r}, t),$$
$$\mathbf{u} = \sum_k \sqrt{\frac{\hbar}{2\rho\omega_k}} \left( b_k e^{i\mathbf{kr} - i\omega_k t} + b_{-\mathbf{k}}^\dagger e^{-i\mathbf{kr} + i\omega_k t} \right), \quad (8)$$

where $\mathbf{u}(\mathbf{r}, t)$ is the lattice displacement vector, $D$ is the deformation potential coupling constant, $\omega_k = s|\mathbf{k}|$ is the phonon frequency, and $\rho$ is the surface mass density of graphene sheet. Plugging these quantities into the Golden Rule for the el-ph emission rate gives

$$d\Gamma = \frac{d\theta}{2\pi} \nu |V_{fi}|^2 \frac{2\pi}{\hbar} \left( N_{ph}(\mathbf{k}) + 1 \right), \quad (9)$$

where $\theta$ is the angle parameterizing the Fermi surface, and the deformation potential matrix element equals $|V_{fi}| = \sqrt{\frac{\hbar}{2\rho\omega_k}} D |\mathbf{k}| \langle \psi_f | \psi_i \rangle$, with the overlap $\langle \psi_f | \psi_i \rangle = \cos(\theta_{\mathbf{p'}} - \theta_{\mathbf{p}})$ accounting for the chirality of charge carriers. Here $\mathbf{p}$ and $\mathbf{p'}$ are electron momenta, and $\mathbf{k} = \mathbf{p} - \mathbf{p'}$. (Parenthetically, for monolayer graphene, the cos factor is to be replaced with $\cos((\theta_{\mathbf{p'}} - \theta_{\mathbf{p}})/2)$.) The density of final states equals $\nu = m^*/(2\pi\hbar^2)$, where $m^* = 0.033 \, m_e$ is the carrier effective mass; since electron–phonon scattering preserves carrier spin and valley index, the relevant degeneracies are not included in $\nu$.

Phonon absorption is described by a similar expression with $N_{ph}(\mathbf{k}) + 1$ replaced by $N_{ph}(\mathbf{k})$. Since temperatures of interest are considerably larger than the Bloch-Gruneisen temperature $T_{BG} = \hbar s k_F$, we can approximate the Bose factors $N_{ph}(\mathbf{k})$ and $N_{ph}(\mathbf{k}) + 1$ as $T/\hbar\omega_k$. Plugging $N_{ph}(\mathbf{k}) + N_{ph}(\mathbf{k}) + 1 \approx 2T/\hbar\omega_k$ in the expression for $d\Gamma$ and replacing $k$ with $\mathbf{p} - \mathbf{p'}$, gives

$$d\Gamma = \cos^2(\theta) \frac{d\theta}{2\pi} \frac{\pi\nu D^2}{\hbar\rho s^2} 2T, \quad (10)$$

**Table 1 Electron-phonon scattering**

| Mean free path $l_{el-ph}$ | SLG | BLG |
|---|---|---|
| $T = 100$ K, $n = 10^{12}$ cm$^{-2}$ | ~6 μm | ~2–4 μm |
| Scaling | $n^{-1/2}/T$ | $n^{1/2}/T$ |

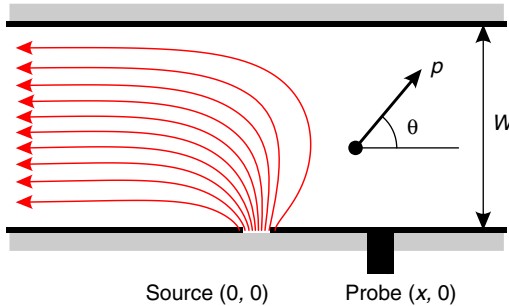

**Fig. 5** The vicinity geometry in a strip of width $w$. The red lines illustrate current injected through the source at $x = 0$ and drained far to the left, at $x = -\infty$. Voltage probe, positioned at a distance $x$ from the source, is used to measure potential $V_p(x)$ relative to the ground far to the right, at $x = +\infty$. The source and drain contacts, as well as the probe, were positioned at the $y = 0$ boundary. The angle $\theta$ between the electron momentum $p$ and the strip edge parameterizes states at the two-dimensional Fermi surface

Then the transport scattering rate equals

$$\Gamma_{tr} = \oint d\Gamma (1 - \cos\theta) = \frac{2\pi\nu D^2 T}{\hbar\rho s^2} \oint \frac{d\theta}{2\pi} \cos^2(\theta)(1 - \cos\theta) = \frac{\pi\nu D^2 T}{\hbar\rho s^2}. \quad (11)$$

The electron–phonon mean free path is given by $l_{el-ph} = \nu/\Gamma_{tr}$, where $\nu = \hbar k_F/m^*$ is the carrier velocity. For bilayer graphene, we assume surface mass density $\rho = 2 \times 7.6 \times 10^{-7}$ kg/m$^2$, the speed of sound $s = 2 \times 10^4$ m/s. In single-layer graphene, transport measurements are consistent with deformation potential $D$ of the order of 20 eV, see, e.g., ref. [41]. For bilayer graphene, ab-initio calculations[42] yield $D = 15$ eV. Assuming $D$ in the range of 15–20 eV, we arrive at $l_{el-ph}$ of the order of 3 μm for typical experimental conditions.

Table 1 provides a summary of the results for the single-layer and bilayer graphene. These estimates are in agreement with the el-ph scattering rates extracted from the temperature dependence of the four-probe resistance reported in the ref. [16].

**Details of the theoretical model.** To describe the ballistic and viscous regimes on equal footing and provide a link between them, we use the kinetic Boltzmann equation for quasiparticles at the Fermi surface. Expanded to linear order in the deviation $\delta f$ from the equilibrium Fermi-Dirac distribution, Boltzmann equation reads

$$\nu \nabla_x \delta f(p, x) - I_{ee}(\delta f(p, x)) = J(p, x). \quad (12)$$

The collision operator $I_{ee}$ in (12) describes scattering between single-particle states via momentum-conserving ee collisions. Near the Fermi surface, the distribution can be parameterized by the standard ansatz $\delta f(p) = -\frac{\partial f_0}{\partial \varepsilon} \chi(p)$, where the energy dependence in $\chi$ can be ignored on the account of fast quasiparticle thermalization by collinear scattering at the 2D Fermi surface[32,33]. We analyze the angular dependence $\chi(\theta)$, where the angle $\theta$ parameterizes the Fermi surface and $\hat{p} = (\cos\theta, \sin\theta)$ is the unit vector along the carrier momentum. We assume that all non-conserved angular harmonics of $\chi(\theta)$ relax with equal rates $\gamma_{ee} = v_F/l_{ee}$, whereas the three angular harmonics corresponding to the conserved net momentum and particle number do not relax. The operator $I_{ee}$, linearized in $\chi$, therefore takes the form[13,22]:

$$I_{ee}(\chi(\theta)) = -\gamma_{ee}(\chi(\theta) - \langle\chi(\theta')\rangle - 2\hat{p} \times \langle\hat{p'}\chi(\theta')\rangle). \quad (13)$$

The angular brackets denote angular averaging over $\theta'$.

To model the current flow in the strip geometry shown in Fig. 5, the Eq. (12) is to be furnished with the boundary conditions describing momentum relaxation at the strip edges. We assume that particles are scattered diffusely, following Lambert's law. Hence the edges $y = 0$ and $y = W$ effectively become isotropic current sources. At $y = 0$, we write

$$\chi(\theta > 0, x) = J(\theta, x) + \frac{1}{2}\int_0^\pi \sin\theta' \chi(-\theta', x)\mathrm{d}\theta'. \qquad (14)$$

The choice of the coefficient 1/2 in the second term is dictated by current conservation. Indeed, for an isotropic distribution of outgoing particles, $\chi(\theta > 0) = \chi_0$, the outgoing particle flux, $\nu \int v_F \sin\theta \chi(\theta)\mathrm{d}\theta/2\pi$, is given by $\nu v_F \chi_0/\pi$. Here $\nu$ is the density of particle states, and $v_F$ is the Fermi velocity. In the absence of current injection, this quantity must be equal to the incoming flux which is given by the integral in the second term. Similarly, an isotropic current source attached to the boundary, $I(x, 0)$, is described via $J(\theta, x) = \pi I(x)/(e\nu v_F)$ in the Eq. (14). For the opposite orientation of the boundary, $y = W$, positive and negative angle values in the Eq. (14) are to be interchanged.

In general, distribution of particles in the Knudsen regime is not represented by a local equilibrium Fermi function, and a local chemical potential cannot be introduced. This poses a difficulty in relating the signal on a probe contact to the distribution function. To resolve this, we adopt the model of a probe which is commonly used to describe leads in mesoscopic circuits, see e.g.,[43]. A probe is a perfect absorber for nonequilibrium carriers, which are equilibrated inside the probe and subsequently re-emitted into the fluid with an isotropic angular distribution. If the open-circuit condition is maintained in the probing circuit, the potential on the probe is proportional to the influx $F$ of charge carriers into the probe,

$$F = \nu \int_0^{-\pi} v_F(-\sin\theta)\chi(\theta, x)\frac{\mathrm{d}\theta}{2\pi}. \qquad (15)$$

Since outgoing charge carriers are in equilibrium with the probe potential $V_p$, they are characterized by the distribution function $\chi = eV_p$, so that the outgoing flux is $\nu e V_p/\pi$. Balancing these fluxes, one finds the probe potential

$$V_p(x) = \frac{1}{2e}\int_0^\pi \sin\theta \chi(-\theta, x)\mathrm{d}\theta. \qquad (16)$$

In the hydrodynamic regime, the distribution function is given by an equilibrium expression, which can be related to the local electric potential, $\chi(\theta, x) \approx e\phi(x)$. In this limit, one finds $V_p(x) = \phi(x)$. For a generic nonequilibrium distribution, however, the relation between the local potential $\phi(x)$ and the probe signal $V_p(x)$ is less straightforward. In particular, in the ballistic limit ($l_{ee} = \infty$) the probe attached to the edge of the sample does not register particles grazing along the edge. This suppresses the space charge effect, however this suppression is not a universal phenomenon, and should be viewed as an approximation.

**Numerical modeling**. To model the experimental geometry of Fig. 1a, we analyze the flow induced in a strip of width $W$, $0 < y < W$ by a point source on its edge at the point $(0, 0)$ and a drain at $x = -\infty$, see Fig. 4. Such a flow can be represented as a superposition of a symmetric flow emitted by the source with a uniform flow directed to the drain electrode. Both flows can be analysed numerically via the approach described below.

First, we pass to the Fourier representation with respect to the coordinate $x$ along the strip, and discretize the transverse coordinate: $y_n = nh$, where $h = W/N_y$ is the step size, $n = 0,\ldots, N_y - 1$. We also discretize the momentum direction as $\theta_i = \pi(i + 1/2)/N_\theta$, $i = 0,\ldots 2N_\theta - 1$. Hence the distribution function becomes a function of the wavevector $k$ and two discrete coordinates,

$$\chi(x, y = nh, \theta = \theta_i) = \int \chi_{n,i}(k)e^{ikx}\frac{\mathrm{d}k}{2\pi}. \qquad (17)$$

We employ the following finite-difference representation of the kinetic equation (13):

$$\frac{\sin\theta_i}{h}\left[\chi_{n,i} - \chi_{n-1,i}e^{-ikh\cot an\theta_i}\right] = I_{ee}[\chi_{n,i'}],$$
$$\frac{\sin\theta_i}{h}\left[\chi_{n+1,i}e^{ikh\cot an\theta_i} - \chi_{n,i}\right] = I_{ee}[\chi_{n,i'}]. \qquad (18)$$

For numerical stability, the scheme is made "upwind": the form on the first line should be applied for upward-going particles ($0 < \theta_i < \pi$), and the form on the second line describes particles propagating downwards. Due to the choice of the exponential factors, the exact solution of (12) in the collisionless limit ($\gamma_{ee} = 0$), $\chi(k, y, \theta) \propto \exp(-iky\cot an\theta)$, satisfies the discretized equation.

Thus, discretization of the advection term in Boltzmann equation, $(v\nabla)\delta f$, links the values of the distribution function at nearby sites. The discretized form of the collision integral (13) mixes propagation angles within the same site. Therefore, the above finite-difference system, together with the boundary conditions (14) can be recast into the well-known three-diagonal form in which only blocks on three adjacent sites $n$ and $n \pm 1$ are coupled:

$$A_{n,ij}(k)\chi_{n-1,j}(k) + B_{n,ij}(k)\chi_{n,j}(k) + C_{n,ij}(k)\chi_{n+1,j}(k) = b_{n,i}(k). \qquad (19)$$

Here $A_{n,ij}(k)$, $B_{n,ij}(k)$ and $C_{n,ij}(k)$ are matrix operators acting on the angular index $i$ describing propagation of particles and scattering between different momentum directions. The right-hand side $b_{n,i}(k)$ describes external sources of particles. Such a system can be efficiently solved via the standard three-diagonal matrix algorithm[44].

The point source was represented as a source term in the Eq. (14), with Fourier image $I(k) = 1$. The uniform Poiseuille-like flow can be obtained by analyzing the $k = 0$ limit of the three-diagonal system (19), in which the flow is dragged by an external bias field. The bias field is incorporated into the Eq. (12) via the term $-eE\cos\theta$. The value of the bias field $E$ is then obtained by normalizing the solution to the total current of 1/2.

To make sure that the details of the boundary layer near the edges are simulated properly, we have chosen a rather fine grid, $N_y = 5000$. The propagation angles were discretized with $N_\theta = 50$, which corresponds to 3.6° step in $\theta$. The particle distribution $\chi_{n,i}(k)$ was calculated for $|k|W < 50$, which gives a satisfactory approximation to the distances of interest, $0.1W < x < W$. The probe signal was then calculated as the particle flux (16), giving the results shown in Figs. 1c, 2b, 3b.

### Data availability

The datasets generated and/or analyzed during the current study are available from the corresponding author on reasonable request.

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

## Acknowledgements

We gratefully acknowledge support from the Simons Center for Geometry and Physics where some of the research was performed. D.A.B. and A.K.G. acknowledge the financial support from Marie Curie program SPINOGRAPH, Leverhulme Trust, the Graphene Flagship and the European Research Council. R.K.K. research was supported by EPSRC Doctoral Prize fellowship. G.F. research was supported by the Minerva Foundation, ISF Grant 882 and the RSF Project 14-22-00259. L.L. acknowledges support of the Center for Integrated Quantum Materials under NSF award DMR-1231319; and Army Research Office Grant W911NF-18-1-0116.

## Author contributions

L.S.L., G.F., and A.K.G. designed and supervised the project. M.B.S. fabricated the devices. Transport measurements and data analysis were carried out by D.A.B., A.I.B. and R.K.K. Theory analysis was done by A.V.S., L.S.L., and G.F. The manuscript was written by A.V.S., D.A.B., L.S.L., A.K.G., and G.F. A.I.B. and I.V.G. provided experimental support. All authors contributed to discussions.

## Additional information

**Competing interests:** The authors declare no competing interests.

