## [Peer Review File · Nature Communications]

Reviewers' comments:

Reviewer #1 (Remarks to the Author):

The authors have written an excellent paper describing the experiment and theory for viscous electron fluids and negative nonlocal resistance across the ballistic-to-hydrodynamic crossover. Subject to a few minor things below, the paper was a compelling and easy read. Well done!

At least in the theoretical model, I'd think that without momentum-relaxing scattering, R_v depends only on l_{ee}/w (up to overall prefactors that set the dimensions, etc.). Here w is a channel width, assuming that other spatial dimensions are negligible. So I would expect that -- if there were no momentum relaxing scattering, or finite size effects, and if the relaxation time kinetic theory is a good model -- the data presented in Figure 3 would all collapse onto a single function after some suitable re-scalings and subtractions of Ohmic effects. Do the authors agree that such a collapse would occur, with the 3 caveats listed above? I think some discussion along these lines in the main text would be very helpful. This is especially true if R_v would always be negative in the absence of momentum-relaxing scattering. Then the w above which R_v becomes negative is set by non-universal disorder physics, and not by hydrodynamics. In contrast, if R_v is positive in the ballistic limit anyway, then the authors' characterization of where hydrodynamics begins and ends seems rather semantic and debatable (hydrodynamics is an asymptotic series in $l_{ee}d/dx$, most likely).

Assuming my understanding as outlined above, is correct, then the way the authors introduce the idea in the abstract may be misleading. They state that physics was "interaction dominated but non-hydrodynamic in a wide temperature range". The authors' phrasing immediately brings to mind the proposal of Ref. [28]: namely, that there is an "in-between hydrodynamics" that is neither conventional Navier-Stokes nor ballistic transport, which should be visible in 2d electron systems. However, the authors use the simplest toy model of the hydro-to-ballistic crossover in this paper, which does not exhibit this "in-between" regime.

I am pretty enthusiastic to recommend publication in Nature Communications. The one thing that makes me hesitate is that this work is in many ways a follow up to earlier work that the authors wrote about in Science a few years ago. I would therefore say that this work is not particularly innovative or novel, but it does give a very improved theoretical/experimental treatment of an existing problem. There is also a theoretical arXiv preprint 1806.09538, with some overlap in the author list, that seems to have some overlap with the theory in the present manuscript. My view is that the problem studied here is important enough that follow up work is still very impactful. So with this in mind, I do encourage the editors to accept this paper after revisions, if they are fine with the relation of this work to other recent work of the authors.

Reviewer #2 (Remarks to the Author):

The manuscript "Probing Maximal Viscous Response of Electron System at the Onset of Fluidity" by D. A. Bandurin, A. V. Shytov, L. S. Levitov, et al. investigates the nonlocal transport in bilayer graphene in the regime that corresponds to the transition from the non-hydrodynamic to hydrodynamic behavior. Such a transition is experimentally demonstrated via the crossover in the vicinity resistance. The experimental results are in a good agreement with the numerical analysis. Provided data is technically sound and agrees with the previous studies. Further, the ballistic to hydrodynamic crossover in the graphene is studied experimentally for the first time. However, the results are rather narrow and cannot be viewed as of extreme importance in the specific field and, probably, are not interesting outside of the condensed matter discipline. Indeed, the crossover from ballistic to hydrodynamic regime is trivially expected as an intermediate state between these already explored regimes. Therefore, while this study is suitable for a good condensed matter journal, I cannot recommend its publication in Nature Communications.

Nevertheless, there are several issues that should be addressed by the authors before any publication:

- 1) The penultimate sentence in the abstract is misleading. While the authors state that the viscous response is strongest just before vanishing, this is not clear from Fig.2, where there is a relatively smooth transition. The corresponding statement should be clarified.
- 2) In the last sentence in the abstract, the authors claim that their manuscript provides "a first demonstration of how fluidity appears and develops in an interacting electron system". However, in some aspects, such an experimental study was already performed in the 3D Weyl semimetal WP₂ in the preprint arXiv:1706.05925. Further, the "appearance of the fluidity" is a rather opaque term that was already clarified before (see item 3). Therefore, the authors should refine the corresponding statement.
- 3) The authors could do a better job in citing the earlier studies. In particular, the possibility of the hydrodynamic transport in solids was predicted for the first time in references R. N. Gurzhi, J. Exp. Theor. Phys. 17, 521 (1963) and R. N. Gurzhi, Sov. Phys. Uspekhi 11, 255 (1968). Further, a hydrodynamic flow of electrons was experimentally studied in a 2D AlAs and GaAs heterostructures in references L. W. Molenkamp and M. J. M. de Jong, Solid State Electron. 37, 551 (1994) and M. J. M. de Jong and L. W. Molenkamp, Phys. Rev. B 51, 13389 (1995). In addition, the electron hydrodynamics in monolayer and bilayer graphene and the so-called hydrodynamic window were considered in paper D.Y.H. Ho, I.Yudhistira, N. Chakraborty, and S. Adam, Phys. Rev. B 97, 121404(R) (2018).
- 4) In the introduction, where the authors discuss the possible hydrodynamic effects in the electron fluids, it is more appropriate to state that they are "predicted" but not "found". Indeed, while the provided references are theoretical studies, the current description might create a false impression that these effects have been already experimentally observed.
- 5) After Eq.(1) the authors state that R_v has a negative sign across the entire transition. However, this is not the case in the data provided in Figs.1, 2, and 3, where R_v is positive at small temperature. Therefore, the corresponding statement should be clarified.
- 6) The acronym BV for quasiballistic-to-hydrodynamic transition is confusing and should be changed.
- 7) There is a typo in the formula for l_{ee} in the bilayer graphene. Instead of $l_{ee} \sim n^{3/2}$ it should read $l_{ee} \sim n$.
- 8) The panels in Fig.2 do not correspond to the caption as well as the description in the text. In particular, while it is stated that Fig.2b corresponds to the experimental data, the caption in the figure states that the results are theoretical. The authors should correct this issue.
- 9) It is not clear why the Poisson-Boltzmann framework, which is used to theoretically model the ballistic/hydrodynamic motion, is called the quantum kinetic equation. Indeed, the approach utilized in the Supplementary information is a usual kinetic theory with an additional equation for the electric potential. Since a similar framework was already used in various studies, the statement in the introductory part that "neither there exist theoretical approaches treating both ballistic and viscous regimes on equal footing, nor any systematic experimental study of the transition has been performed" is misleading and should be clarified.
- 10) There is a typo on 7 page of the manuscript. The current is drained on the far left $x=-\infty$ not $y=-\infty$.
- 11) Since the theoretical and experimental results look different in the panels in Fig.2, the statement that Figs.2a and 2b are practically identical is misleading and should be either clarified or removed.
- 12) There is a typo in the Fig.3 caption. The units for the electrons number density are cm^{-2} not cm^2 .
- 13) The manuscript will clearly benefit from including a plot or insert showing the Knudsen number. This will simplify the corresponding discussion at the end of the manuscript.
- 14) Next, there are few questions regarding the theoretical framework. Since the authors consider sufficiently large temperatures, the phononic contribution to the scattering and the thermoelectric contributions to the current should be estimated. The corresponding hydrodynamic window should be also discussed. Next, the hydrodynamic currents could generate a magnetic field, which is not taken into account in the proposed framework. While such an effect could be small, it is still worth

discussing or estimating.

15) Further, in Supplementary information the authors use both capital and small letter w for the strip width. Since small w is used in the main text, the capital w should be replaced with a small one.

Reply to Reviewer 1

The authors have written an excellent paper describing the experiment and theory for viscous electron fluids and negative nonlocal resistance across the ballistic-to-hydrodynamic crossover. Subject to a few minor things below, the paper was a compelling and easy read. Well done! We thank the reviewer for careful reading of our manuscript and a kind assessment.

At least in the theoretical model, I'd think that without momentum-relaxing scattering, R_v depends only on l_{ee}/w (up to overall prefactors that set the dimensions, etc.). Here w is a channel width, assuming that other spatial dimensions are negligible. So I would expect that -- if there were no momentum relaxing scattering, or finite size effects, and if the relaxation time kinetic theory is a good model -- the data presented in Figure 3 would all collapse onto a single function after some suitable re-scalings and subtractions of Ohmic effects. Do the authors agree that such a collapse would occur, with the 3 caveats listed above?

In the absence of momentum relaxation, the quantity R_v is a function of three lengths, l_{ee} , w and x (distance from the source to the point of measurement). Temperature dependences presented in fig 3 can indeed be plotted as functions of l_{ee}/w . However, different curves correspond to different x/w and they cannot be collapsed by vertical axis rescaling because x -dependences are different in the ballistic and hydrodynamic regimes. As referee stated correctly, non-universality appears because of finite-size effects represented in this case by the scale w . Particularly, while in the ballistic regime R_v is negative at $x \ll w$, reflection from the opposite wall in the given geometry makes R_v positive at x comparable to w . Only in the limit $w \gg x, l_{ee}$, the dependences on x , T and n can be all collapsed into universal dependence $R_v(l_{ee}/x)$, which indeed will be negative everywhere (absent Ohmic terms as shown in arXiv preprint 1806.09538 that the referee mentions and which we cite in the revised version of this manuscript).

I think some discussion along these lines in the main text would be very helpful. This is especially true if R_v would always be negative in the absence of momentum-relaxing scattering. Then the w above which R_v becomes negative is set by non-universal disorder physics, and not by hydrodynamics. In contrast, if R_v is positive in the ballistic limit anyway, then the authors' characterization of where hydrodynamics begins and ends seems rather semantic and debatable (hydrodynamics is an asymptotic series in $l_{ee}d/dx$, most likely).

Referee is right that the crucial point is whether R_v can be negative in the quasi-ballistic regime, where l_{ee} is of larger or comparable to the sample's dimensions. The answer to this is affirmative, as we show in this paper experimentally and theoretically (and add more details to the theory in the archive article mentioned). We also agree with the Referee about the relation between hydrodynamics and the asymptotic series in $l_{ee}d/dx$. In physical terms, the smallness of this parameter means that a) the ee collision mean free path l_{ee} is shorter than the lengthscale x at which the system being probed, and b) that the momentum non-conserving scattering is negligible at that scale, and c) that the nonlocal signal decays with temperature. This is precisely the regime that we access in our system at higher temperatures ($l_{ee} < x$), i.e. when the vicinity resistances passes through its minimum. Hence our identification of the viscous regime and its onset at the minimum of $R_v(T)$ is in line with the referee's comments, which we found most helpful for substantially rewriting the manuscript along these lines. We believe the paper is much clearer now.

Assuming my understanding as outlined above, is correct, then the way the authors introduce the idea in the abstract may be misleading. They state that physics was "interaction dominated but non-hydrodynamic in a wide temperature range". The authors' phrasing immediately brings to mind the proposal of Ref. [28]: namely, that there is an "in-between hydrodynamics" that is neither conventional Navier-Stokes nor ballistic transport, which should be visible in 2d electron systems. However, the authors use the simplest toy model of the hydro-to-ballistic crossover in this paper, which does not exhibit this "in-between" regime.

We do not mean here "in-between" regime of ref 28. When we call the low-T side of the resistance minimum "interaction-dominated but a non-hydrodynamic regime", we mean that the transport is dominated by the interactions but the behavior is qualitatively different from a hydrodynamic regime. This occurs when l_{ee} is greater than the distance to the probe but shorter than w , so that ee collisions happen on the scale shorter than the transport time across the hall bar. In this case, as discussed in the revised manuscript, interactions dominate R_V (making it negative), but since the distance x at which the system is being probed is shorter than l_{ee} , the response is non-hydrodynamic. This leads to temperature and doping dependences of R_V which are distinct from that expected in the hydrodynamic regime.

I am pretty enthusiastic to recommend publication in Nature Communications. The one thing that makes me hesitate is that this work is in many ways a follow up to earlier work that the authors wrote about in Science a few years ago. I would therefore say that this work is not particularly innovative or novel, but it does give a very improved theoretical/experimental treatment of an existing problem. There is also a theoretical arXiv preprint 1806.09538, with some overlap in the author list, that seems to have some overlap with the theory in the present manuscript. My view is that the problem studied here is important enough that follow up work is still very impactful. So with this in mind, I do encourage the editors to accept this paper after revisions, if they are fine with the relation of this work to other recent work of the authors.

We are grateful to the referee for appreciation of our work. Further to it, let us note that this work is not just some incremental addition to the Science2016 article (and others), it sheds a new light on the basic premise of the previous works that negative resistance is a smoking gun of viscous regime. In the revised version we commented specifically on the findings in the Science2016 article. We show in the present work theoretically and experimentally (and add more theoretical details in the archive article mentioned) that negative signal already exists in the quasi-ballistic regime, i.e. where electron-electron mean free path l_{ee} is comparable with the system size. In our geometry, we l_{ee} predict and observe the reduction of positive, purely ballistic, signal upon increasing the temperature from that of liquid nitrogen where l_{ee} is much larger than the sample size w and the measurement position x . This observation is rather intriguing as, primarily, the reduction of ballistic signal is always routinely associated with momentum-non-conserving scattering. In this work we describe the transition between purely ballistic ($l_{ee} > x, w$) to hydrodynamic ($l_{ee} < x, w$) via interaction-facilitated quasi-ballistic regime ($x < l_{ee} < w$). Furthermore, we suggest in this paper a completely new way to identify the transition to the viscous regime and show experimentally that it works. This new way is related to the seemingly paradoxical statement that viscosity is low in the extreme hydro regime and grows higher as we increase l_{ee} and the system is becoming more ballistic. This non-trivial picture indeed describes the behavior of the measured voltage and thus allows to identify the boundary between regimes as negative resistance maximum (rather than zero). We believe that this novelty is quite dramatic.

Reply to Reviewer 2

The manuscript "Probing Maximal Viscous Response of Electron System at the Onset of Fluidity" by D. A. Bandurin, A. V. Shytov, L. S. Levitov, et al. investigates the nonlocal transport in bilayer graphene in the regime that corresponds to the transition from the non-hydrodynamic to hydrodynamic behavior. Such a transition is experimentally demonstrated via the crossover in the vicinity resistance. The experimental results are in a good agreement with the numerical analysis. Provided data is technically sound and agrees with the previous studies. Further, the ballistic to hydrodynamic crossover in the graphene is studied experimentally for the first time. However, the results are rather narrow and cannot be viewed as of extreme importance in the specific field and, probably, are not interesting outside of the condensed matter discipline. Indeed, the crossover from ballistic to hydrodynamic regime is trivially expected as an intermediate state between these already explored regimes. Therefore, while this study is suitable for a good condensed matter journal, I cannot recommend its publication in Nature Communications.

While it may indeed be trivial to expect a crossover between the free-particle regime and the hydrodynamic regime, our main finding is a very different one. Namely, we find a non-monotonic dependence of the electric response on temperature and show that this behavior is due to a completely different crossover, not anticipated before. This crossover is between two different interaction-dominated regimes, the hydrodynamic regime in which the ee interactions dominate completely, and the "quasi-ballistic" regime in which l_{ee} is larger than the distance to the probe but smaller or comparable with the system size. In the latter case the observed T and n dependences of the response are strikingly different from the prediction of both hydrodynamic and free-particle models, and hence indicate a new, non-trivial, interaction-dominated but not hydrodynamic regime (see also the discussion in the reply to the Reviewer 1).

Moreover, our analysis demonstrates that merely observing a negative $R_V(T)$ does not provide a convincing signature of the hydrodynamic regime, as was previously stipulated in the literature. Instead, it can occur well outside the hydrodynamic regime, when $l_{ee} \ll l_{ph}$.

Understanding the precise nature of the crossover and the new regime is also quite important for ongoing research: in fact, most systems in which the signatures of viscosity were reported were essentially in the crossover region that we discuss in our paper. Therefore, the most basic question posed by the existing data is as follows: whether the boundary between ballistic and viscous regimes corresponds to nonlocal resistance changing sign or having a maximum? To answer this question, we needed to develop understanding of both regimes and the formalism applicable in both. On the viscous side, we arrive at a seemingly paradoxical statement that viscosity is low in the extreme hydro regime and grows higher as we increase l_{ee} and the system is becoming more ballistic. On the quasi-ballistic side, we show that resistance can be negative as well if l_{ee} is comparable or larger than the sample dimensions. This is so because interaction effects can dominate outside of the hydrodynamic regime, as evident through the negative R_V , which decreases in magnitude as we increase l_{ee} . That non-trivial picture indeed describes the behavior of the measured voltage, allowing us to identify the onset of fluidity and crossover to hydrodynamics as the negative resistance maximum (rather than zero). We substantially rewrote the text to make this clear.

Nevertheless, there are several issues that should be addressed by the authors before any publication:

- 1) The penultimate sentence in the abstract is misleading. While the authors state that the viscous response is strongest just before vanishing, this is not clear from Fig.2, where there is a relatively smooth transition. The corresponding statement should be clarified.*

Color map of Fig 2 does not allow one to appreciate the shape of this dependence. However, it is perfectly visible in the curves of Fig 1, particularly sharp in the right panel. As one can see, lowering temperature one monotonically increases negative resistance (on the interval of some 150K) until it reaches maximum (which we identify as boundary between quasi-ballistic and viscous regimes), then fast drops to zero (on the interval of less than 10K). We believe this factor 15 justifies “strongest just before vanishing” statement.

- 2) In the last sentence in the abstract, the authors claim that their manuscript provides "a first demonstration of how fluidity appears and develops in an interacting electron system". However, in some aspects, such an experimental study was already performed in the 3D Weyl semimetal WP_2 in the preprint arXiv:1706.05925. Further, the "appearance of the fluidity" is a rather opaque term that was already clarified before (see item 3). Therefore, the authors should refine the corresponding statement.*

It is of course true that signatures of the hydrodynamic behavior have been reported in previous experiments on a few different systems. Instead, here we provide a detailed characterization of the transition between the ballistic and hydrodynamic behavior. As explained in detail above, this transition is driven by interactions, and has a rather unexpected character not anticipated in previous literature. Thus our work represents a substantial addition to existing knowledge. In the revised version we explained in detail what we mean by appearance of fluidity and what are the defining properties of the electric response in the fluid state (particularly, temperature dependence).

- 3) The authors could do a better job in citing the earlier studies. In particular, the possibility of the hydrodynamic transport in solids was predicted for the first time in references R. N. Gurzhi, J. Exp. Theor. Phys. 17, 521 (1963) and R. N. Gurzhi, Sov. Phys. Uspekhi 11, 255 (1968). Further, a hydrodynamic flow of electrons was experimentally studied in a 2D AlAs and GaAs heterostructures in references L. W. Molenkamp and M. J. M. de Jong, Solid State Electron. 37, 551 (1994) and M. J. M. de Jong and L. W. Molenkamp, Phys. Rev. B 51, 13389 (1995). In addition, the electron hydrodynamics in monolayer and bilayer graphene and the so-called hydrodynamic window were considered in paper D.Y.H. Ho, I.Yudhistira, N. Chakraborty, and S. Adam, Phys. Rev. B 97, 121404(R) (2018).*

We thank the reviewer for pointing to our poor reference list. In the revised manuscript we include missing references.

- 4) In the introduction, where the authors discuss the possible hydrodynamic effects in the electron fluids, it is more appropriate to state that they are "predicted" but not "found". Indeed, while the provided references are theoretical studies, the current description might create a false impression that these effects have been already experimentally observed.*

We agree with the reviewer that some of the effects, such as electron vortices, which were mentioned in the introductory section, remain to be experimentally detected. However, most of the listed effects were probed experimentally as apparent from the corresponding citations. Nevertheless, we acknowledge the reviewer’s remark and rewrote this sentence.

5) After Eq.(1) the authors state that R_V has a negative sign across the entire transition. However, this is not the case in the data provided in Figs.1, 2, and 3, where R_V is positive at small temperature. Therefore, the corresponding statement should be clarified.

We state that R_V is negative across the transition, but not across the entire range under discussion. Indeed, R_V is positive at liquid He temperatures, signaling ballistic transport. However, in the vicinity of the transition point, where $R_V(T)$ trend changes from descending to transcending, the sign of R_V is negative. This is again related to our main statement that the transition corresponds to R_V minimum.

6) The acronym BV for quasiballistic-to-hydrodynamic transition is confusing and should be changed.

We thank the reviewer for pointing to the confusion arising due to this acronym. In the revised manuscript we avoid using this notation.

7) There is a typo in the formula for l_{ee} in the bilayer graphene. Instead of $l_{ee} \sim n^{3/2}$ it should read $l_{ee} \sim n$.

According to Landau Fermi-liquid theory, the e-e scattering rate is given by $\gamma_{ee} = (k_B T)^2 / \hbar E_F$. The corresponding mean free path, therefore, is expressed as $l_{ee} = v_F / \gamma_{ee}$. In BLG, $E_F = \hbar^2 \pi n / 2 m$ where $m \approx 0.033 m_e$ for experimentally relevant range of carrier densities and $v_F = \hbar \sqrt{\pi n} / m$ that results in $l_{ee} n^{3/2}$.

8) The panels in Fig.2 do not correspond to the caption as well as the description in the text. In particular, while it is stated that Fig.2b corresponds to the experimental data, the caption in the figure states that the results are theoretical. The authors should correct this issue.

We thank the reviewer for pointing to this inaccuracy. We revised the manuscript accordingly.

9) It is not clear why the Poisson-Boltzmann framework, which is used to theoretically model the ballistic/hydrodynamic motion, is called the quantum kinetic equation. Indeed, the approach utilized in the Supplementary information is a usual kinetic theory with an additional equation for the electric potential. Since a similar framework was already used in various studies, the statement in the introductory part that "neither there exist theoretical approaches treating both ballistic and viscous regimes on equal footing, nor any systematic experimental study of the transition has been performed" is misleading and should be clarified.

Our main finding, as discussed in detail above, is that the ballistic-to-viscous transition is driven by interactions, and that there exists quasi-ballistic regime which is not a free-particle regime but is dominated by interactions. We describe this new behavior using the Poisson-Boltzmann framework which accounts properly for the fermion exclusion through Pauli blocking of ee scattering in the Fermi sea, a scheme that is conventionally called the quantum-Boltzmann approach. We are pleased to see that this well-known approach is capable of describing a new regime that has not been mapped out in the previous literature. We agree with the referee that the statement needed clarification, so we rewrote it in the revised version.

10) There is a typo on 7 page of the manuscript. The current is drained on the far left $x=-\infty$ not $y=-\infty$.

We thank the reviewer for pointing to this inaccuracy. We revised the manuscript accordingly.

11) Since the theoretical and experimental results look different in the panels in Fig.2, the statement that Figs.2a and 2b are practically identical is misleading and should be either clarified or removed.

In the manuscript we emphasize that the maps are “practically identical up to a rescaling of the T axis” which is indeed the case. In our model, the e-e collision rate γ_{ee} , being the only T -dependent parameter in the approach, is taken from Landau Fermi-liquid theory only for instructive purposes and the model itself can be applied to any other functional dependence of γ_{ee} on T and n . Note that in the next paragraph, we provide a detailed explanation why a simple Landau Fermi-liquid scattering rate is incapable to describe e-e collisions in graphene and its bilayer accurately.

12) There is a typo in the Fig.3 caption. The units for the electrons number density are cm^{-2} not cm^2 .

We are grateful to the reviewer and apologize for this typo. We correct the revised manuscript.

13) The manuscript will clearly benefit from including a plot or insert showing the Knudsen number. This will simplify the corresponding discussion at the end of the manuscript.

We thank the reviewer for this suggestion. We have revised the Figure 3 and included corresponding Knudsen number on the top-axis.

14) Next, there are few questions regarding the theoretical framework. Since the authors consider sufficiently large temperatures, the phononic contribution to the scattering and the thermoelectric contributions to the current should be estimated. The corresponding hydrodynamic window should be also discussed. Next, the hydrodynamic currents could generate a magnetic field, which is not taken into account in the proposed framework. While such an effect could be small, it is still worth discussing or estimating.

Indeed, at elevated temperatures, electron-phonon scattering significantly impacts electron transport. However, in the present work, as we state in the experimental part, we subtract all contributions to the vicinity resistance stemming from momentum-non-conserving scattering events, including phonons, following the approach we have developed in our previous works (Science. 351, 1055–1058 (2016), Nat. Phys. 13, 1182–1185 (2017)) and that was found to be a fair analysis procedure.

We can also neglect thermoelectric contributions since we work far from charge neutrality point and apply very small currents that do not cause significant heating of the electronic temperature. That is, we work in the regime linear with respect to current. Likewise, the magnetic field generated by such current can be neglected – there is no need to add respective discussion since field generation in the hydrodynamic regime has been calculated before and reported in Phys. Rev. B. 92, 165433 (2015).

15) Further, in Supplementary information the authors use both capital and small letter w for the strip width. Since small w is used in the main text, the capital w should be replaced with a small one.

We thank the reviewer for pointing to our inaccuracy. We revised the draft accordingly.

REVIEWERS' COMMENTS:

Reviewer #1 (Remarks to the Author):

I think the authors have done a good job in improving the manuscript in response to the first round of review.

A careful understanding of the crossover between two well understood regimes is important to do properly. As the authors have now made much more clear, realistic experimental devices may exhibit multiple length scales and the hydrodynamic crossover can be much more subtle than a simple handwaving argument would suggest. It is also worth emphasizing that hydrodynamics in 'quantum' fluids has been a subject of broad appeal throughout physics, including in nuclear physics, condensed matter and cold atoms over the past decade. I view this paper as worthy of publication in Nature Communications, in present form.

Reviewer #2 (Remarks to the Author):

I find the answers to my questions and the corresponding corrections in the manuscript partially unsatisfactory. Indeed, the key statements regarding the novelty of the results remain mostly unchanged. The latter include, for example, the emergence of the fluid behavior and the theory able to describe both ballistic and viscous regime. In addition, the new title does not properly reflect the content of the paper, which is devoted to the ballistic-hydrodynamic crossover. Thus, I believe that the manuscript "Searching for the Viscous Electron Fluid in Graphene" by D. A. Bandurin, A. V. Shytov, L. S. Levitov, et al. should not be published in Nature Communications at least in its present form. Of course, this study is suitable for a good condensed matter journal.